# Artesunate Switches Monocytes to an Inflammatory Phenotype with the Ability to Kill Leukemic Cells

**DOI:** 10.3390/ijms22020608

**Published:** 2021-01-09

**Authors:** Rubia Isler Mancuso, Sara Teresinha Olalla Saad, Juliana Hofstätter Azambuja

**Affiliations:** Hematology and Transfusion Medicine Center, University of Campinas, Campinas 13083-970, Brazil; rubiaimancuso@gmail.com (R.I.M.); sara@unicamp.br (S.T.O.S.)

**Keywords:** artesunate, hematologic malignancies, immunotherapy, monocytes

## Abstract

Monocytes are components of the tumor microenvironment related to cancer progression and immune escape. Therapeutic strategies for reprogramming monocytes from a tumor-supporting phenotype towards a tumoricidal phenotype are of great interest. Artesunate (ART) may be an interesting option for cancer treatment; however, the role of ART in regulating the inflammatory tumor microenvironment has not yet been investigated. Our aim is to evaluate the immunomodulatory potential of ART in vitro in human primary monocytes. ART treatment induced an increase in inflammatory monocytes (CD14^high^CD16^−^) with HLA-DR high expression and MCP-1/IL-1β release. On the other hand, ART treatment reduced CD206 and CD163 expression, and abolished the monocyte population known as non-classical and intermediate. Leukemia cells in contact with monocytes programmed with ART presented enhanced in vitro apoptosis suggesting that monocytes acquired the ability to kill leukemic cells. ART induced changes in the monocyte phenotype were mediated by JAK2/STAT3 downregulation. The induction of immunosuppressive environment is an important step for cancer progression. ART showed an immunomodulatory activity, leading immune cells to an antitumor phenotype and could be a candidate for immunotherapy in cancer patients.

## 1. Introduction

Hematologic malignancies account for 10% of all annual deaths due to cancer [1]. Despite the poor prognosis of hematologic malignancies, the treatment of these disorders has remained largely unchanged over the past several decades. Currently, high dose chemotherapy with several adverse effects remains as the main therapy [2]. This has created an impetus to explore novel therapeutic approaches, such as immunotherapies [3].

Hematological tumor cells live in an immune cell-enriched microenvironment, unique and substantially different from that of other solid tumors [3,4]. Monocytes belong to this tumor microenvironment (TME) and after a tumor-induced “immunoediting,” these monocytes, which are originally the first line of defense against tumor cells, undergo a phenotypic switch and become tumor-supportive and immunosuppressive [5,6]. Accordingly, repolarizing leukemia-associated monocytes with more M1-like inflammatory characteristics eliminate their pro-leukemic effects and reduce tumor progression [7]. Thus, monocyte plasticity highlights the reprogramming of monocytes as an attractive therapeutic strategy to inhibit tumor progression, enabling these cells to adapt their function to meet the needs of antitumor defense [3]. Therapies inducing systemic immune activation may also have the capability of reprograming monocytes before they arrive to the tumor site, probably exerting influence on their tumor infiltration activity.

Artemisinin is a semi-synthetic compound of the sesquiterpene lactone drug family, and is obtained from a Chinese plant, *Artemisia annua* L., used for centuries in traditional Chinese medicine and known for its antimalarial properties [8]. Artesunate (ART) is a more stable and soluble derivative of artemisinin that has been shown to exert several pharmacological actions such as anti-inflammatory [9,10,11], anti-leishmanial [12,13], neuroprotective [14,15], immunomodulatory [16], and antitumoral activity [17]. However, the effects of ART in reprogramming monocytes from a tumor-supporting phenotype into an antitumor phenotype remain unexplored. Our objective was to investigate the potential of ART as a monocyte reprogramming agent.

## 2. Results

### 2.1. ART Is Safe for Human Primary Monocytes

We first determined the ART cytotoxic profile by exposing monocytes cultures to ART for 24 h. ART did not induce monocytes cytotoxicity in a range between 25–5000 μM (IC_50_ = 2205 μM). These results indicated that the concentrations 100–500 µM were safe for monocytes (Figure 1A). We next examined whether ART induced apoptosis of primary monocytes. Monocytes treated with 100, 200, and 500 μM of ART did not show a significant increase in apoptosis (Figure 1B,C). Based on this, we chose non-cytotoxic concentrations (100, 200, and 500 μM) to perform the next experiments (Figure 1D). In addition, monocytes showed no morphological changes after the treatment (Appendix A).

### 2.2. ART Induces Monocytes Phenotypic Changes to a Pro-Inflammatory Phenotype

Currently, the circulating human monocytic cells can be separated into three main subsets: (i) CD14^high^CD16^−^ (classical monocytes), (ii) CD14^high^CD16^+^ (intermediate monocytes), and (iii) CD14^−/low^CD16^+^ (non-classical monocytes). In the prevention protocol, classical monocytes increased after LPS-treatment (from 19.35 ± 6.55 to 63.45 ± 8.15, *p* = 0.0398). Naive monocytes treated with increased doses of ART also increased CD14^high^CD16^−^ population in a dose-dependent manner (24.4 ± 11.7, 40.45 ± 15.55, and 74 ± 2, respectively, compared to M0). ART showed similar results in IL-4-treated monocytes (33.15 ± 2.95, 44.1 ± 2.9, and 71.95 ± 1.75, respectively) (Figure 2A,B).

Intermediate monocytes decreased after LPS-treatment (from 67.9 ± 5.6 to 5.26 ± 1.24, *p* = 0.0195). A reduction in intermediate monocyte population after ART-treatment was observed in a dose-dependent manner, with a significant difference at 500 μM (from 67.9 ± 5.6 to 10.87 ± 7.43, *p* = 0.0339). A similar result was observed in IL-4-treated monocytes with ART (Figure 2A,B). Furthermore, non-classical monocytes decreased with LPS-treatment (from 4.895 ± 0.395 to 0.94 ± 0.22, *p* = 0.0255) and increased after IL-4-treatment (from 4.895 ± 0.395 to 8.915 ± 0.735, *p* = 0.0231). ART-treated monocytes reduced in a dose-dependent manner (7.925 ± 0.305, 5.555 ± 0.905, and 1.395 ± 0.445, respectively compared to M2). The same pattern was observed in the reversal protocol, (Figure 2C,D). Taken together, ART induced monocytes phenotypic changes to an inflammatory phenotype.

As expected, IL-4 treated monocytes increased CD206 in the prevention protocol, (from 8.37 ± 0.4661 to 54.37 ± 1.087, *p* < 0.0001) (Figure 3A,B). Naive monocytes treated with ART reduced CD206 expression at 500 μM (from 8.37 ± 0.4661 to 2.267 ± 0.4083, *p* = 0.0065). Furthermore, IL-4 ART-treated monocytes also reduced CD206 expression (41.87 ± 0.7219, 32.87 ± 0.6566, and 15.9 ± 0.1528, respectively compared to M2). The same pattern was observed with the CD163. Monocytes treated with IL-4 and ART reduced CD163 expression at 500 μM (from 40.73 ± 7.891 to 11.25 ± 0.15, *p* = 0.0181).

In contrast, LPS-treated monocytes increased HLA-DR (from 4.975 ± 0.175 to 8.15 ± 0.63, *p* = 0.0022). Naive ART-treated monocytes increased HLA-DR at 500 μM (from 4.975 ± 0.175 to 7.555 ± 0.185, *p* = 0.0093). IL-4 ART-treated monocytes increased HLA-DR (7.44 ± 0.27 and 7.83 ± 0.11, 200 and 500 μM respectively compared with M2). No difference was observed in CD80 after ART treatment. Similar patterns were observed in the reversal protocol (Figure 3C,D).

In the prevention protocol, increased levels of NO were observed in LPS-treated monocytes (from 95.6 ± 0.7 to 113.5 ± 1.5, *p* < 0.0001) and a decrease in NO was observed in IL-4 treated monocytes (from 95.6 ± 0.7 to 26.35 ± 1.25, *p* < 0.0001). However, ART was able to increase NO production in IL-4 treated monocytes (26.0 ± 1.9, 31.4 ± 0.5, and 37.3 ± 1.8, respectively) (Figure 4A). The reversal protocol showed similar results (Figure 4B).

### 2.3. ART Induces Pro-Inflammatory Cytokine Release from Monocytes

IL-4 treated monocytes showed reduced levels of MCP-1 (from 1529 ± 255.8 to 40.67 ± 11.75, *p* = 0.0016) and IL-1β (from 639 ± 68.57 to 369.2 ± 29.91, *p* = 0.6653) compared to basal levels (M0). Furthermore, ART reverts this immunosuppressive switch by increasing MCP-1 release (from 40.67 ± 11.75 to 275.1 ± 31.16, *p* = 0.0002) (Figure 4C,D) and IL-1β (from 369.2 ± 29.91 to 1195 ± 101.1, *p* < 0.0001) (Figure 4E,F). In addition, ART treatment was not able to modulate TNF-α, RANTES, INF-γ or IL-8 release (Figure 4G,H). Taken together, our data indicate that ART treatment induces an M1-like phenotype similar to that induced by LPS treatment. In addition, ART is capable of converting naive and immunosuppressive monocytes to this inflammatory phenotype.

### 2.4. ART Downregulates JAK2/STAT3 Pathway

Western Blot analyses were performed to evaluate the possible pathways related with ART immunomodulatory activity. A reduction of 7.1, 4.1, and 3.5 times was observed in p-JAK2 when naive monocytes were treated with increased doses of ART (100, 200, and 500 μM, respectively). Moreover, IL-4 treated monocytes showed a reduction of 11.1, 7.1, and 3.0 times in p-JAK2 compared with M2-control (Figure 5A). In addition, p-STAT3 increased four times in IL-4 monocytes compared with M0. Moreover, ART reduced in 4.0, 7.0, and 2.9 times p-STAT3 in IL-4 treated monocytes compared with M2-control (Figure 5B). No changes in NF-kB, IKKα/β, p-JNK, p-c-Jun, ERK, p-p38, p-eIF2, CHOP, or ATF4 were observed (Figure 5C).

### 2.5. ART Induces a Tumoricidal Phenotype

To study the apoptotic effects of ART, monocytes previously treated with ART were co-cultured with U937, HL60, and OCI-AML3 cell lines (Figure 6A). U937 cell apoptosis increased after co-culture with monocytes ART-programmed (from 4.97 ± 0.3135 to 22.0 ± 2.373, *p* < 0.0001). In addition, an increase of U937 apoptotic cells was observed in IL-4 treated monocytes with ART (from 7.13 ± 0.5045 to 33.0 ± 1.0, *p* < 0.0001) (Figure 6B,C). The same pattern was observed with HL-60 and OCI-AML3 cell lines (Figure 6C). Taken together, these data indicate that ART induces monocytes to assume an attack phenotype, characterized by a tumoricidal activity and ability to kill leukemic cells in vitro.

## 3. Discussion

The present study demonstrates that ART reduces in vitro leukemia growth by modulating monocytes activation state to an antitumor phenotype. First, we observed that ART was effective in inducing a strong inflammatory monocyte polarization, mainly characterized by high MCP-1 and IL-1β release, HLA-DR expression, NO production, and low CD206/CD163 expression, which, in turn, promoted an antitumoral phenotype and induced in vitro U937, HL-60 and OCI-AML3 apoptosis. This monocyte switch is in part mediated by a decrease in the phosphorylation of JAK2/STAT3. We propose that ART and the consequent induction of an inflammatory monocyte phenotype may revert the tumor-induced blockade of immune surveillance, which permits antitumor responses and tumor regression.

Cancer is a public health problem. Conventional cancer therapy, such as chemotherapy, provides limited efficacy with side effects and drug resistance. Thus, there is an urgent need for novel drugs which target cancer cells specifically, without side effects. ART emerges as a good option for cancer treatment with IC values ranging from 0.092 to 1.5 μM in leukemic cell lines [17,18]. Here, we show that the IC_50_ for ART in healthy primary immune cells is 2205 μM demonstrating that ART is extremely selective and non-cytotoxic for healthy cells.

Antitumor action of ARTs mainly involves induction of apoptotic cell death, ROS generation, and cell cycle arrest [17,18,19,20]. Nevertheless, our understanding of the ART/mechanisms is far from complete. Therefore, a few evidences indicate that ARTs antitumor affects may be related with immunoregulatory activities and cancer metabolism changes mainly by decreasing the secretion of immunosuppressive cytokines (TGF-β/IL-10) and the infiltration of inhibitory immune cells such as TRegs and MDScS, while increasing the infiltration of attack cells, T CD8+ and NK with increased antitumor potential [16,21,22,23,24,25]. However, the immunomodulatory effect on monocytes/macrophages remains unclear.

Monocytes are a heterogeneous subset of cells with distinct subpopulations that can assume several phenotypes and secrete a variety of substances capable of impacting physiological processes or the development of diseases [6,26]. Recently, monocytes have emerged as important regulators of cancer progression and aggressiveness, with different phenotype subtypes appearing to have opposing impacts on tumor growth metastatic spread [6]. The most widely reported cancer-induced phenotypic alteration in circulating monocytes is the switch to an immunosuppressive and tumor friendly phenotype [27,28]. Thus, the identification of therapies with the ability of “re-educating” monocytes to a tumoricidal phenotype will most likely represent a useful therapy strategy for patients with cancer.

The present work demonstrates that ART modulates monocyte polarization, causing an increase in classical monocytes and completely abolishing intermediate and non-classical monocytes. Interestingly, classical monocytes are described as important cells during inflammatory and infectious processes, appearing to be critical effectors during the early phase of the antitumor response, in particular by killing tumor cells [6]. Our data show that monocytes treated for 24 h with ART assume an antitumor phenotype and induce apoptosis of leukemic cells. This suggests that ART is capable of leading monocytes to an antitumor phenotype and this phenotype remains even after the drug is withdrawn. In a clinical context, this may indicate that the effect of ART therapy may persist to eliminate the tumor via the reprogramming of immune cells to an inflammatory and tumor-killing phenotype. In addition, tumors often escape antitumor immune responses through critical immune checkpoint molecules. Thus, the immunomodulatory potential of ART herein described could be a new strategy to create synergy with immune checkpoint inhibitors.

Interestingly, ART also alters the cytokine release profile by increasing the release of MCP-1 and IL-1β. Both are important players in building a defensive immune response. The main activity of MCP-1 is to attract blood monocytes and activate monocytes for tumor cell killing phenotype [29]. In addition, IL-1β acts as an amplifier of immune reactions inducing MCP-1 expression in monocytes, regulating myeloid cell recruitment into tumor tissue, and leading to inflammation [30]. Therefore, both cytokines are important for the recruitment and activation of monocytes in the tumor site, allowing the assembly of an anti-tumor immune response, as well as the activation and maturation of other immune cells present in the TME.

The activation of JAK2/STAT3 is known as an important pathway related to IL-4 response in immune cells. Collectively, the present results suggest that inhibition of the JAK2/STAT3 pathway may be a potential mechanism, via which ART mediates monocyte switch to an inflammatory phenotype. Notably, STAT3 is an emergent target anti-cancer therapy [31], and data have indicated that STAT3 inhibition in immune cells may reprogram these cells to a tumoricidal phenotype, reduce tumor cell proliferation, angiogenesis, and metastasis [32,33]. The inhibition of JAK2/STAT3 activities in the monocytes programmed by ART may explain the switch to an antitumor phenotype in these monocytes. In addition, these data suggest a new mechanism of action for a previously established drug.

In conclusion, our data indicate that ART treatment induces monocytes to an inflammatory and antitumor phenotype (Figure 7), which may reduce tumor recurrence and progression. However, future studies are required to better understand the long-term effects of ART on the modulation of immune cell phenotypes.

## 4. Materials and Methods

### 4.1. Monocytes

Peripheral blood mononuclear cells were isolated from buffy coats of healthy blood donors by Ficoll-Paque (17-1440-03, GE Healthcare Bioscience, Chicago, IL, USA) density gradient centrifugation. Monocytes were isolated by Percoll (17-5445-02, GE Healthcare Bioscience, Chicago, IL, USA) density gradient centrifugation followed by adherence to a plastic dish for 1 h as previously described [34]. All subjects donating blood specimens signed an informed consent approved by the Ethical Committee of University of Campinas Hospital number 1.242.880 approved on 09-24-2015.

### 4.2. Monocytes Treatment

ART was dissolved in DMSO and subsequently mixed with RPMI with 10% FBS to obtain solutions at final concentrations (DMSO final concentration 0.3%). To induce changes compatible with an immunosuppressive phenotype, in order to test the potential of ART in reprogramming monocytes, we used two different strategies, namely prevention and reversal protocols (Figure 1D,E). In the reversal protocol, the cells were first exposed to IL-4 (20 ng/mL, PeproTech Roch, city, state, country) for 6 h aiming to program monocytes into an immunosuppressive phenotype (M2-like) and were subsequently washed and treated in a fresh medium with ART (Sigma-Aldrich Corp., St. Louis, MO, USA) for an extra 18 h. In the prevention protocol, the potential of the ART was tested by blocking the immunosuppressive monocytes formation, the cells were exposed to different concentrations of ART and immediately treated with IL-4 (20 ng/mL) for 24 h. The control naive cells M0 were maintained in RPMI with 10% FBS and the M1-like positive cells were treated with LPS (10 ng/mL, Sigma-Aldrich Corp., St. Louis, MO, USA).

### 4.3. Cell Death and Viability Assays

#### 4.3.1. MTT Assay

Cell viability was determined using the (4,5-dimethyl)-2,5diphenyltetrazolium bromide assay (MTT) assay, which consists of measuring the number of cells with metabolically active mitochondria based on the reduction of the tetrazolium salt MTT to formazan. Briefly, monocytes (2 × 10^5^ cells/well) were seeded in 96-well plates. After 1 h, cultures were replaced with fresh media in the absence or presence of increasing ART concentrations (25–5000 μM). Following 24 h of treatment, 0.5 mg/mL MTT were added and incubated for 3 h. Subsequently, 0.1 N HCl was added to anhydrous isopropanol to solubilize the formazan. The absorbance was determined in a microplate reader at 570 nm. Results were expressed as percentage of control. Cultures with the same volume of DMSO in RPMI 10% served as controls.

#### 4.3.2. Apoptosis Assay

Monocytes (1 × 10^6^ cells/well in a 24-well plate) were treated with ART for 24 h. Cells were washed with PBS and resuspended in a binding buffer containing 1 mg/mL propidium iodide (PI) and 1 mg/mL APC-labeled annexin V for 15 min at room temperature (RT) protected from light. 10.000 events were analyzed in the FACS Calibur (BD Bioscience, Franklin Lakes, NJ, USA).

### 4.4. Activation Phenotype Markers

#### 4.4.1. Flow Cytometry

##### Membrane Markers

Phenotypic analysis was performed using monoclonal antibodies against CD14, CD80, CD206, CD16, CD163, and human leukocyte antigen D related (HLA-DR) and the monocytes population was gated in CD14 positive cells. For this, after treatment as described in item 2.2, cells were suspended in the staining buffer with the Fluorochrome-conjugated antibodies listed in Appendix A and incubated for 30 min at RT. After washing at least 2 times in the flow cytometry buffer, cells were immediately analyzed using an FACS Calibur (BD Bioscience, Franklin Lakes, NJ, USA).

##### Nitric Oxide Production

Intracellular NO generation was measured by flow cytometry following staining with 4-amino-5-methylamino-2′,7′-difluorofluorescein diacetate (Sigma-Aldrich Corp., St. Louis, Missouri, USA). Acquisition of cells was performed on a FACS Calibur flow cytometer and analysis was carried out using the FlowJo software (v.10).

#### 4.4.2. ELISA

TNF-α, IL-1β, IL-8, MCP-1, and RANTES levels were determined in the supernatant of monocytes cultures by ELISA, following the manufacturer’s recommendations (R&D Systems, Minneapolis, MN, USA). Results were expressed as pg cytokine/mL based on a standard curve.

### 4.5. Western Blot

After 15 min of treatment, total cell protein was extracted with RIPA buffer. Protein concentrations were quantified by Bradford method [35]. Equal protein amounts (30–40 µg) were loaded on 8 to 10% SDS polyacrylamide gels and electrophoretically transferred to nitrocellulose membranes. Membranes were then blocked by incubation with 5% fat-free dry milk, followed by an overnight incubation with a specific primary antibody (Appendix A). One hour of RT incubation was then carried out using horseradish peroxidase-conjugated secondary antibody. The immunoreactivities were visualized by ECL Western Blot Analysis System (Amersham Pharmacia Biotech, UK).

### 4.6. Apoptotic Activity of ART

#### 4.6.1. Tumor Cell Lines and Co Culture System

U937, HL-60, and OCI-AML3 cell lines were purchased from the American Type Culture Collection (ATCC) (Manassas, VR, USA) and were cytogenetically tested and authenticated before being frozen. Cells were grown in RPMI, IMDM, and -MEM, respectively, supplemented with 10% (*v*/*v*) (FBS) at 37 °C and in the atmosphere of 5% CO_2_ in air. Monocytes were seeded in 12 multi-well plates at densities of 2 × 10^6^ cells/well and treated for 24 h. Next, cells were washed at least 2 times and for co-culture purpose, 2 × 10^5^ U937 or HL60 or OCI-AML3 cells (1:10 target/effector cells ratio) were seeded in the same well and allowed to communicate overnight.

#### 4.6.2. Apoptosis Assay

Monocytes were co-cultured overnight with PKH26-labeled target cells (U937, HL-60, or OCI-AML3) in fresh medium with 10% FBS. The cells were then resuspended in Annexin-V-binding buffer (BD Bioscience, Franklin Lakes, NJ, USA) containing Annexin-V-APC. After 15 min incubation at RT, samples were analyzed with a FACS Calibur flow cytometer. For apoptosis cells were separated in PKH26+ (tumor) and PKH26− (monocytes) and percent of Annexin V were determined.

### 4.7. Statistical Analysis

Data analysis were performed using GraphPad Prism 7 software. Results were expressed as mean ± SEM and subjected to one-way analysis of variance followed by Tukey–Kramer post hoc test and considered different at * *p* < 0.05.

## Figures and Tables

**Figure 1 ijms-22-00608-f001:**
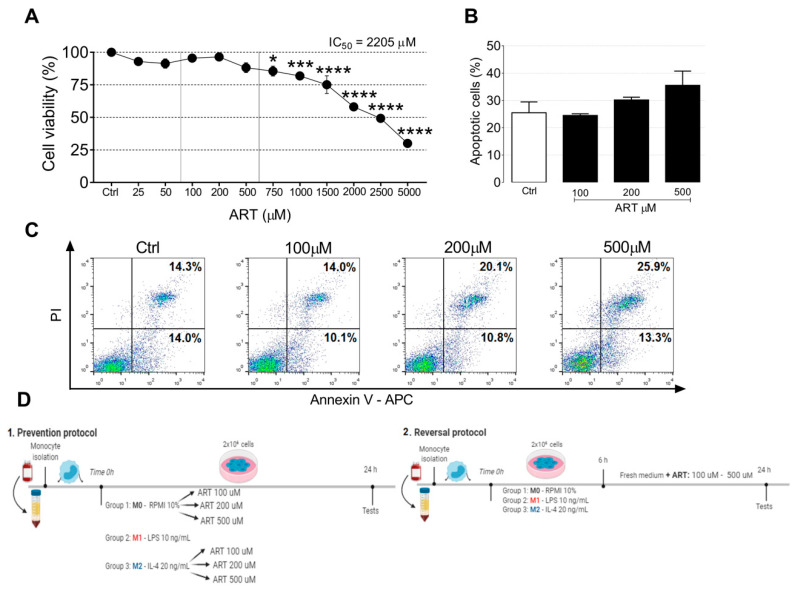
ART effects on human primary monocytes cell viability. Cell viability of monocytes treated with increasing doses of artesunate (ART) (25–5000 µM) for 24 h measured by MTT assay (**A**). Cell viability of cells treated with DMSO (vehicle) were considered 100%. *, *** and, **** indicate significant difference from the control group as determined by ANOVA followed by Tukey post hoc (*p* < 0.05, 0.001, and <0.0001, respectively). Monocytes were treated with 100, 200, and 500 µM of ART for 24 h. After treatment, the cells were stained with Annexin V/PI (**B**,**C**). Experimental design of the two protocols used in the study (**D**). Monocytes were obtained from buffy coats of healthy blood donors. In the prevention protocol, cells were treated with LPS for M1-like monocytes or IL-4 for M2-like monocytes and ART were added at the same time. In the reversal protocol, cells were treated with LPS and IL-4 for 6 h and then ART was added in fresh medium for an additional 18 h. Cells were then collected for the next experiments.

**Figure 2 ijms-22-00608-f002:**
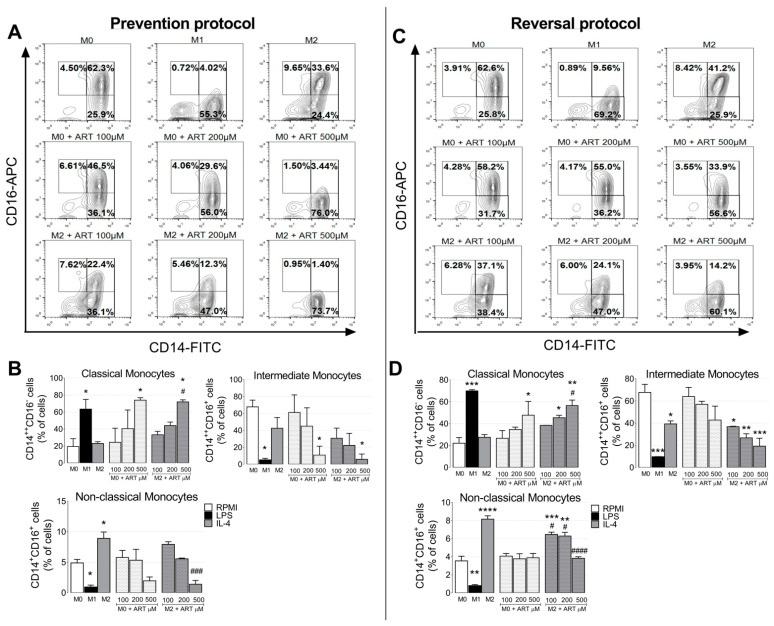
ART treatment induces an inflammatory phenotype of monocytes. In the prevention protocol, monocytes were treated with increasing doses of ART (100, 200, and, 500 µM) for 24 h in the presence or not of IL-4 (**A**,**B**). In the reversal protocol (**C**,**D**), cells were treated with IL-4 for 6 h and ART was then added in fresh medium for an extra 18 h. Expression of cell surface markers were analyzed by flow cytometry, and cells were divided into three main subsets: (**i**) CD14^high^CD16^−^ (classical monocytes), (**ii**) CD14^high^CD16^+^ (intermediate monocytes), and (**iii**) CD14^−/low^CD16^+^ (non-classical monocytes). Data represent mean ± SEM of at least three independent experiments performed in triplicate. Data were analyzed by ANOVA followed by Tukey post hoc. *, **, ***, and, ****, significantly different from control M0 cells (white bar) (*p* < 0.05, <0.01, <0.001, and <0.0001, respectively). #, ###, and, ####, significantly different from M2 cells (grey bar). M0: Naive control monocytes maintained in RPMI 10%, M1: positive control for inflammatory monocytes maintained in LPS and, M2: immunosuppressive monocytes maintained in IL-4.

**Figure 3 ijms-22-00608-f003:**
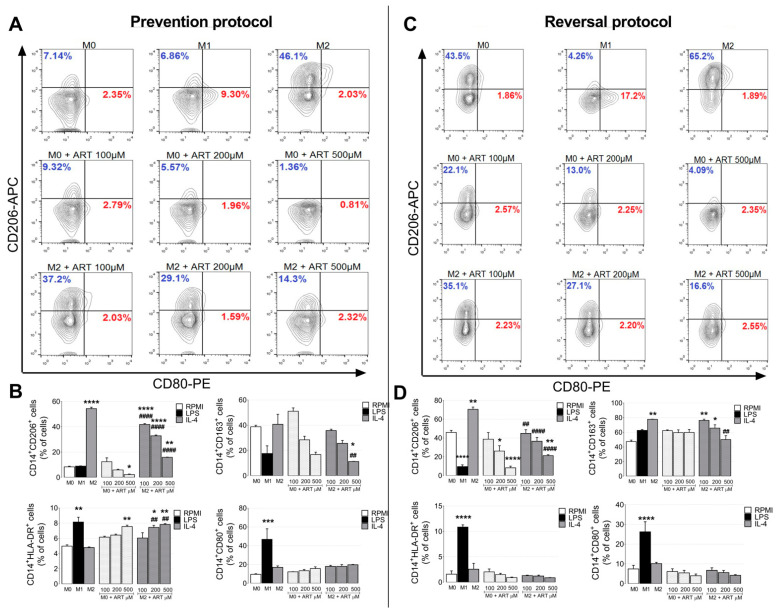
ART treatment induces phenotypic changes of human primary monocytes. Monocytes were treated with increased doses of ART (100, 200, and, 500 µM) for 24 h in the prevention protocol in the presence or not of IL-4 (**A**,**B**). In the reversal protocol (**C**,**D**), cells were treated with IL-4 for 6 h and ART was then added in fresh medium for an additional 18 h. Phenotypic markers for M1- and M2-like monocytes (CD80, CD206, CD163, and HLA-DR) were analyzed by flow cytometry. Populations were gated for CD14 positive cells. Red percentages represent CD80 positive cells and blue percentages CD206 positive cells. Data represent mean ± SEM of at least three independent experiments performed in triplicate. Data were analyzed by ANOVA followed by Tukey post hoc. *, **, ***, and ****, significantly different from control M0 cells (white bar). ## and ####, significantly different from M2 cells (grey bar). M0: Naive control monocytes maintained in RPMI 10%, M1: positive control for inflammatory monocytes maintained in LPS and, M2: immunosuppressive monocytes maintained in IL-4.

**Figure 4 ijms-22-00608-f004:**
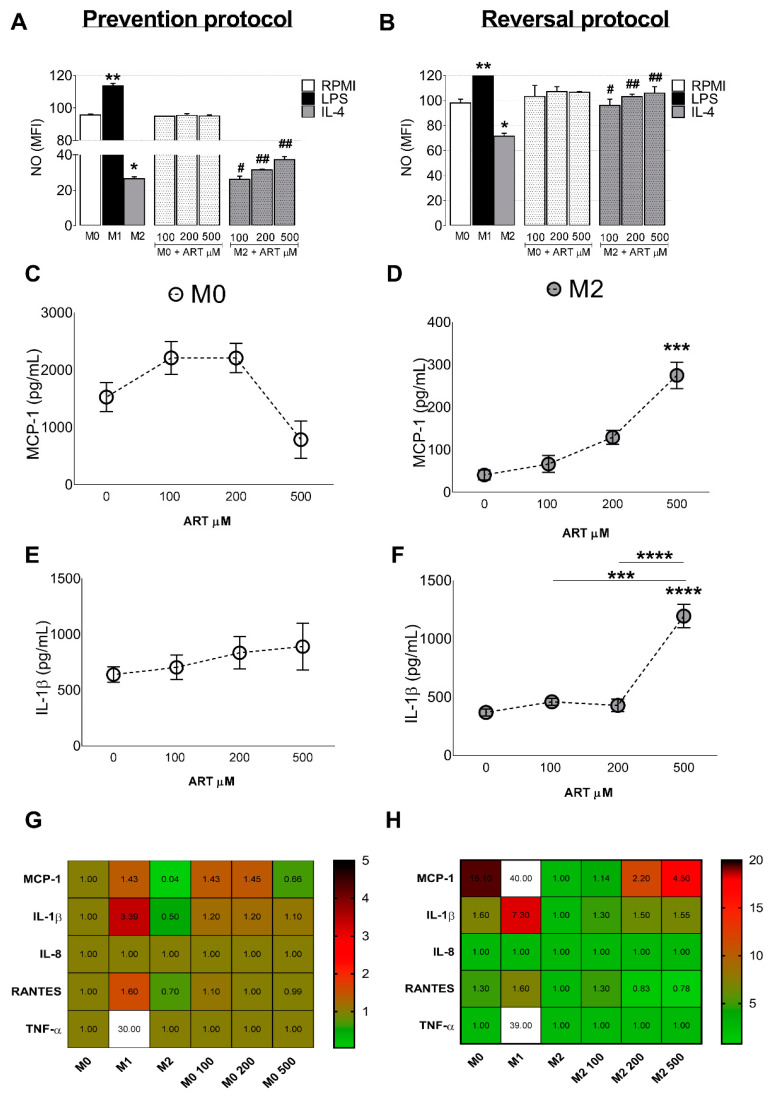
ART treatment induces NO production and MCP-1/IL-1β release. Monocytes were treated with increased doses of ART (100, 200, and 500 µM) for 24 h in the prevention protocol in the presence or not of IL-4 (**A**). In the reversal protocol (**B**), cells were treated with IL-4 for 6 h and ART was then added in fresh medium for extra 18 h. NO production was measured by flow cytometry. Cytokine MCP-1 (**C**,**D**) and IL-1β (**E**,**F**) production was measured at the supernatant of cell culture by ELISA. Heat map showing fold change of all cytokines analyzed (**G**,**H**). Data represent mean ± SEM of at least three independent experiments performed in triplicate. Data were analyzed by ANOVA followed by Tukey post hoc. *, **, *** and, **** significantly different from control M0 cells (white bar). # and ##, significantly different from M2 cells (grey bar). M0: Naive control monocytes maintained in RPMI 10%, M1: positive control for inflammatory monocytes maintained in LPS and, M2: immunosuppressive monocytes maintained in IL-4.

**Figure 5 ijms-22-00608-f005:**
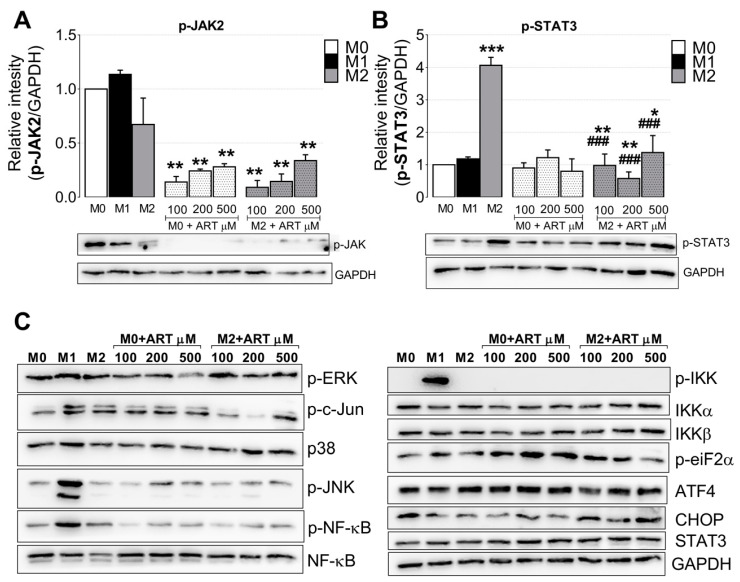
ART downregulates JAK2/STAT3 pathway in IL-4 treated human primary monocytes. Monocytes were treated with increasing doses of ART (100, 200, and 500 µM) for 15 min as in the prevention protocol, in the presence or not of IL-4. Protein was extracted with RIPA buffer and immunoblotted with anti-p-JAK2 (**A**), anti-p-STAT3 (**B**) antibodies, and submitted to densitometry. (**C**) Immunoblotting of the same extracts using antibodies against proteins of the MAPK, endoplasmic reticulum stress, and NF-kB pathway. Data of the densitometric analysis represent mean ± SEM of at least three independent experiments performed in triplicate. Data were analyzed by ANOVA followed by Tukey post hoc. *, ** and ***, significantly different from control M0 cells (white bar). ###, significantly different from M2 cells (grey bar). Images of immunoblotting are representative of one experiment.

**Figure 6 ijms-22-00608-f006:**
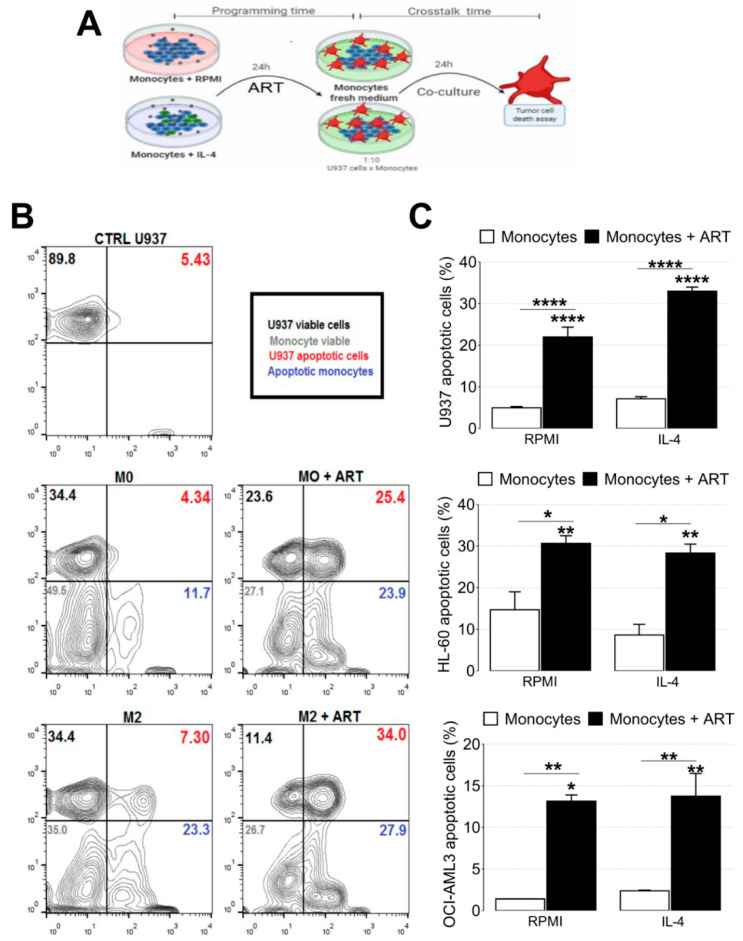
ART treatment reduces in vitro leukemia growth by modulating monocytes with the ability to kill leukemic cells. Experimental design of the co-culture protocol (**A**). Monocytes were treated with 500 µM of ART for 24 h in the presence or not of IL-4. Representative dot plot (**B**). U937, HL60, and OCI-AML3 were added in a fresh medium for more 24 h and the cells were stained with Annexin V (**C**). Data represent mean ± SEM of at least three independent experiments performed in triplicate. Data were analyzed by ANOVA followed by Tukey post hoc. *, **, and ****, significantly different from control RPMI cells or IL-4 cells.

**Figure 7 ijms-22-00608-f007:**
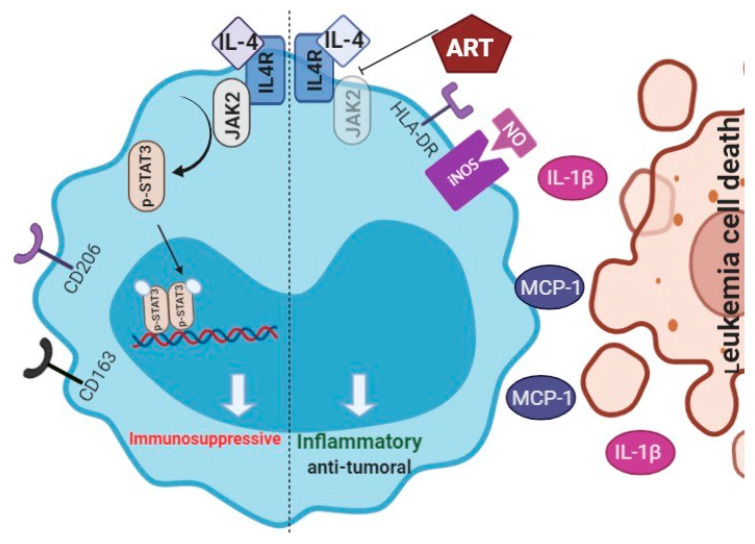
Hypothetic model of ART induction of monocyte phenotypic changes. In an immunosuppressive environment, JAK2/STAT3 are activated after IL-4 release, which increases CD206 and CD163 expression, and decreases MCP-1/IL-1β release and NO production. In contrast, ART inhibits activation of JAK2/STAT3 pathway, increasing HLA-DR expression, NO production, and MCP-1/IL-1β release, and leads to an accumulation of monocytes with an inflammatory profile. In return, CD206, CD163, and CD16 expressions decrease. In co-culture with leukemic cells, monocytes-ART-programmed assumed a tumoricidal profile and induces apoptosis of leukemic cell. These data suggest that ART switches monocytes to an inflammatory and antitumoral profile that could be useful for control tumor progression and re-educates monocytes inside of a tumor microenvironment.

## Data Availability

Raquel S. Foglio participated in writing assistance and English revision; Irene Santos give flow cytometry support; and Mary Ann Foglio participated with the conception of the basic idea.

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
