# Peer review of "Artesunate Switches Monocytes to an Inflammatory Phenotype with the Ability to Kill Leukemic Cells"

_ijms, 2021, doi:10.3390/ijms22020608_

Round 1
Reviewer 1 Report
ART shifts monocyte differentiation to the classical, inflammatory, phenotype instead of intermediate and non-classical, mediated by changes in JAK/STAT signaling. This could be useful to reprogram monocytes to behave as anti-tumor therapy. In this work the authors underwent the characterization of ART-treated monocytes.
The work is well organized and written and methods are clearly described.
My main concern is that authors claim and anti tumor activity of ART-treated monocytes based in coculturing with leukemic cultured lines. I would be more prudent and say (also in the title) that they induce apoptotic activity in cultured cells. Tumors are more complex and authors have not used animal models to confirm that these reprogrammed monocytes actually have an anti-tumor activity
The characterization of the inflammatory phenotype is poor. I would study other chemokine receptors (CXC3R1, CCR7, CXCR4, CCR5) and MAP kinases (K1,K2 etc) as known intermediates
Minor points. The way the results are presented can be clearly improved, e.g.:
-Figure 1C is difficult to read. I assume that the figures inside cytometry diagrams indicate the percentage of apoptotic cells. This should be clearly indicated in the legend and the sign “%” added to the diagrams.
-The same for figures 2 and 3, these should be re-drawn bigger than the originals presented (2B,3B) to make figures readable. Rewrite coordinates in Figure 5B.
Author Response
Reference: Manuscript ijms-1040223 “Artesunate switches monocytes to an inflammatory phenotype with the ability to kill leukemic cells”
We are sending our revised manuscript ijms-1040223. We would like to thank the reviewers for their comments and helpful suggestions, we have addresses all issues, please find our response below.
We would like to make a change the order of the authors and nominate Juliana Hofstätter Azambuja as the correspondent author as her participation was key in this publication.
We hope that the paper is now suitable for publication in the International Journal of Molecular Sciences.
Thank you very much for your kind attention.
Sincerely,
Answer to Reviewer 1
We would like to thank the reviewer for the careful analysis and relevant suggestions that have no doubt improved our work.
Comments:
- 1. My main concern is that authors claim and anti-tumor activity of ART-treated monocytes based in coculturing with leukemic cultured lines. I would be more prudent and say (also in the title) that they induce apoptotic activity in cultured cells. Tumors are more complex and authors have not used animal models to confirm that these reprogrammed monocytes actually have an anti-tumor activity.
We have changed the title to Artesunate switches monocytes to an inflammatory phenotype with the ability to kill leukemic cells” and we have also modified parts of the manuscript in order to avoid the term “antitumor”.
- The characterization of the inflammatory phenotype is poor. I would study other chemokine receptors (CXC3R1, CCR7, CXCR4, CCR5) and MAP kinases (K1,K2 etc) as known intermediates.
Our question was whether artesunate was capable of switching the phenotype of the monocytes, therefore we are most confident that the response is yes, as we measured TNF-α, IL-1β, IL-8, MCP-1, and RANTES, and differences in IL-1β and MCP-1 were clearly observed. Furthermore, we showed surface markers that confirm phenotype, such as CD16, CD206, CD163, CD80, and HLA-DR. Thus, as the results appeared to be so clear to us, we did not consider extra markers in vitro at this point.
Minor points:
- Figure 1C is difficult to read. I assume that the figures inside cytometry diagrams indicate the percentage of apoptotic cells. This should be clearly indicated in the legend and the sign “%” added to the diagrams.
- The same for figures 2 and 3, these should be re-drawn bigger than the originals presented (2B,3B) to make figures readable. Rewrite coordinates in Figure 5B.
We would like to thank the reviewers for calling our attention to these points. We have re-drawn Figures 1, 2, and 3 using bigger letters and numbers. We have also added the sign % to the figures and have re-written the coordinates in Fig 5.
Reviewer 2 Report
On my opinion all experiments are very logical and well done.The most interesting finding is formation of immunomodulated cells after
treatment. That can be used in the immunotherapy.
This is an interesting manuscript and results will be helpful to the researchers in the field of cancer research, as well for the medicinal chemist.
Author Response
Reference: Manuscript ijms-1040223 “Artesunate switches monocytes to an inflammatory phenotype with the ability to kill leukemic cells”
We are sending our revised manuscript ijms-1040223. We would like to thank the reviewers for their comments and helpful suggestions, we have addresses all issues, please find our response below.
We would like to make a change the order of the authors and nominate Juliana Hofstätter Azambuja as the correspondent author as her participation was key in this publication.
We hope that the paper is now suitable for publication in the International Journal of Molecular Sciences.
Thank you very much for your kind attention.
Sincerely,
Reviewer 2
No comments.
Round 2
Reviewer 1 Report
Authors have addressed my concerns and the current version is clearly improved.
Author Response
Thank you very much for the comments and helpful suggestions.